# Fungemia in Hospitalized Adult Patients with Hematological Malignancies: Epidemiology and Risk Factors

**DOI:** 10.3390/jof9040400

**Published:** 2023-03-24

**Authors:** Luz Alejandra Vargas-Espíndola, Sonia I. Cuervo-Maldonado, José L. Enciso-Olivera, Julio C. Gómez-Rincón, Leydy Jiménez-Cetina, Ricardo Sánchez-Pedraza, Katherine García-Guzmán, María José López-Mora, Carlos A. Álvarez-Moreno, Jorge Alberto Cortés, Javier R. Garzón-Herazo, Samuel Martínez-Vernaza, Claudia R. Sierra-Parada, Bryan A. Murillo-Sarmiento

**Affiliations:** 1Facultad de Medicina, Universidad Nacional de Colombia, Bogota 111321, Colombia; 2Infectious Diseases Group, Instituto Nacional de Cancerología, Bogota 111511, Colombia; 3GREICAH—Grupo de Investigacion Enfermedades Infecciosas en Cáncer y Alteraciones Hematológicas, Bogotá 111321, Colombia; 4Microbiology Laboratory, Instituto Nacional de Cancerología, Bogota 111511, Colombia; 5Clínica de Marly, Bogota 110231, Colombia; 6Fundación Sanitas, Clínica Colombia, Bogotá 111221, Colombia; 7Hospital Universitario San Ignacio, Bogotá 110231, Colombia; 8Laboratorio Clínico y de Patología, Clínica Colsanitas, Grupo Keralty, Bogotá 111221, Colombia

**Keywords:** *Candida*, fungemia, hematologic malignancy, candidemia, leukemia, lymphoma, myeloma, mortality

## Abstract

Fungemia in hematologic malignancies (HM) has high mortality. This is a retrospective cohort of adult patients with HM and fungemia between 2012 and 2019 in institutions of Bogotá, Colombia. The epidemiological, clinical, and microbiological characteristics are described, and risk factors related to mortality are analyzed. One hundred five patients with a mean age of 48 years (SD 19.0) were identified, 45% with acute leukemia and 37% with lymphomas. In 42%, the HM was relapsed/refractory, 82% ECOG > 3, and 35% received antifungal prophylaxis; 57% were in neutropenia, with an average duration of 21.8 days. In 86 (82%) patients, *Candida* spp. was identified, and other yeasts in 18%. The most frequent of the isolates were non-albicans *Candida* (61%), *C. tropicalis* (28%), *C. parapsilosis* (17%), and *C. krusei* (12%). The overall 30-day mortality was 50%. The survival probability at day 30 in patients with leukemia vs. lymphoma/multiple myeloma (MM0 group was 59% (95% CI 46–76) and 41% (95% CI 29–58), *p* = 0.03, respectively. Patients with lymphoma or MM (HR 1.72; 95% CI 0.58–2.03) and ICU admission (HR 3.08; 95% CI 1.12–3.74) were associated with mortality. In conclusion, in patients with HM, non-albicans *Candida* species are the most frequent, and high mortality was identified; moreover, lymphoma or MM and ICU admission were predictors of mortality.

## 1. Introduction

The incidence of invasive fungal infections has increased worldwide. More immunosuppressive treatments expose patients to a higher risk of infection and the development of severe disease caused by fungal agents that are considered saprophytes, particularly in debilitated patients and patients treated with chemotherapy for hematological malignancies (HMs) [1,2]. Yeasts are the leading cause of fungemia, and *Candida* spp. is the most common agent, with a reported incidence of 0.15–1.5% in hospitalized patients with underlying malignancies [3]. The reported incidence is three to ten times higher in Latin America [4,5], particularly in Colombia, a middle-income tropical country with limited data on cancer patients. 

The extended use of antifungals for prophylactic and empiric treatment has led to a change in the epidemiology of *Candida* spp. fungemia, with more cases related to non-albicans *Candida*, which are usually azole- and echinocandin-resistant [1,3,6]. The mortality of fungemia in patients with HMs is high (at least 40% in the yeast and 70% in the mold infections), so identification and proper treatment are of the utmost importance. 

The isolated yeast species have regional and populational variations. A prospective trial in eight countries between 2005 and 2009, including patients with HM, found that the proportion of infections with albicans and non-albicans *Candida* was similar (40.4% and 46.5%, respectively). In patients with HMs and allogeneic stem cell transplants, most infections were caused by *C. krusei* and *C. tropicalis*, in contrast with patients with solid tumors where more infections were caused by *C. glabrata* [3]. The most common isolates in candidemia in Latin America after *C. albicans* are *C. parapsilosis* and *C. tropicalis*, and the frequency of *C. glabrata* and *C. krusei* fungemia is lower here than in the United States [7].

Considering the epidemiological variation and the limited information about fungemia in patients with HM in Latin America, in addition to the change in the antifungal susceptibility and the related high mortality, knowledge regarding the epidemiology and risk factors for fungemia in this population is of great interest. 

The objective of this study is to describe the epidemiological, clinical, and microbiological characteristics as well as antifungal susceptibility profile, and to evaluate the independent predictors of mortality in patients with HMs who presented with fungemias between 2012 and 2019 in reference cancer centers in Bogotá, Colombia. 

## 2. Materials and Methods

### 2.1. Scenario

This study was conducted in five reference institutions for cancer treatment in Bogotá, the capital of Colombia, which has a population of more than 8 million inhabitants. The participating institutions were the Instituto Nacional de Cancerología (INC), a public hospital, and Clínica de Marly; Hospital Universitario San Ignacio (HUSI); Clínica Universitaria Colombia; and Clínica Reina Sofía, all of which are private institutions. All these hospitals have oncology or hemato-oncology care units; one of them is public (INC) and all serve patients belonging to both public and private health care providers.

### 2.2. Study Design

A retrospective cohort study was performed. The medical records and microbiological reports of adult patients (aged 18 years or over) with HM and fungemia were reviewed from 1 January 2012 to 31 December 2019 at the participating institutions. The inclusion criteria considered individuals of any race, with confirmed HMs, with or without treatment (chemotherapy, radiotherapy), and at any diagnostic stage. Patients with HIV (human immunodeficiency virus) were excluded. Flowchart of patient selection in Appendix A.

Variables of interest, such as demographic, clinical, and mortality data at 30 days after diagnosis of fungemia, were included. These were collected on REDCap (Vanderbilt University, Nashville, TN, USA), and data analysis was carried out using the STATA (v. 15.0, College Station, TX, USA) and R (v. X, Vienna, Austria) programs.

### 2.3. Definitions

Fungemia was defined as at least one positive blood culture for a strain of yeast or mold (non-*Aspergillus*) species associated with symptoms of fungal infection. Breakthrough fungemia was defined as fungemia that occurred during exposure to an antifungal drug regardless of whether the treatment intent was prophylactic, empiric, preventative, or targeted [8]. Fungemia associated with an intravascular device was considered if the isolated strain from a blood culture taken from a central venous catheter (CVC) grew faster than the peripheral blood (>2 h apart) or in a catheter-tip culture. The decision to remove the CVC was made accordance with IDSA recommendations in neutropenic patients [1]. It was considered an individualized decision in a risk–benefit context taking into account the most likely origin of the candidemia (e.g., CVC-related or gastrointestinal, due to the presence of mucositis) and the risk of catheter removal (bleeding associated with thrombocytopenia and loss of central device).

Four groups of diagnostic stages of hematologic malignancy were defined: the active stage (newly diagnosed/induction), composed of patients with HM who were in induction without an evaluating response or who were in the first cycle of chemotherapy; the maintenance/consolidation stage; the relapsed/refractory stage; and in the disease remission stage.

Neutropenia was defined as an absolute neutrophil count (ANC) < 500 cells/mm^3^ or an ANC that was expected to decrease to <500 cells/mm^3^ during the next 48 h. Severe neutropenia was defined as ANC < 100 cells/mm^3^ [9]. Recovery from neutropenia was defined as a stable ANC above 500 cells/mm^3^ [9]. Early antifungal therapy was defined as the initiation of treatment less than 48 h after the microbiological finding (identification of fungal isolates using Gram-stain in blood). The antimicrobial susceptibility testing methods used were microdilution and CLSI, and the cut-off points were determined by an automatized microbiology system (Vitek, Biomérieux, Marcy L’Etoile, France). The antifungal prophylaxis was indicated according to the institutional protocol of the health institutions based on the 2018 guidelines for antimicrobial prophylaxis from the American Society of Clinical Oncology and the Infectious Diseases Society of America (IDSA) for patients with cancer-related immunosuppression [9], where antifungal prophylaxis was carried out with an oral triazole or parenteral echinocandin. This is recommended for patients at risk of profound and prolonged neutropenia, such as most patients with acute myeloid leukemia/mylodysplastic syndromes or hematopoietic progenitor cell transplant. A total of 32 patients received prophylaxis, 2 with posaconazole and 30 with fluconazole. The median duration of antifungal usage was 16.6 days.

The risk factors were defined as having neutropenia (ANC < 500 cells/mm^3^) for more than 10 days, receiving an allogeneic hematopoietic progenitor cell transplant, taking corticosteroids at doses of ≥0.3 mg/kg/day for >3 weeks, taking anti-T lymphocyte immunosuppressants such as cyclosporine, anti-TNF, alemtuzumab, or nucleoside analogs during the previous 3 months, and severe congenital immunodeficiency.

### 2.4. Statistical Analysis

Data were entered into the REDCap platform, and their statistical analysis was carried out using the R program (version 4.0.6).

The results are expressed as means and standard deviations (SD), as medians and interquartile ranges (IQR) (continuous variables) according to their distribution, or as percentages of the group (categorical variables). Survival was evaluated 30 days after the microbiological finding. To identify the independent variables associated with the outcome, a Cox regression was performed, incorporating the variables that were significant in the univariate model. The assumption of risk proportionality was validated using the Schoenfeld residual test. To select the final model, the backward stepwise elimination method was used, with the Akaike information criterion (AIC) as the elimination criterion.

### 2.5. Ethical Aspects

This project is in line with international regulations, such as the Declaration of Helsinki, the Nuremberg Code, the Belmont Report, and the guidelines established by the Council of International Organizations of Medical Sciences (CIOMS). The conduct of the study was in accordance with good clinical practice (GCP) standards. The level of ethical risk at which the study is classified, according to Article 11 of Resolution No. 008430 of 1993 issued by the Ministry of Health of Colombia, is “Research with minimum risk”. To guarantee confidentiality for the patients included in the research, the variables related to their identity were suppressed (Statutory Law 1581 of 2012 and National Decree 1377 of 2013). The protocol was submitted for review and approval by the ethics committees of the participating institutions. Informed consent was not required by any of the participating institutions since no intervention was performed. 

## 3. Results

### 3.1. Characteristics of the Population

The total number of blood cultures reviewed during the study period (2012–2019) was 6965 blood cultures from 2793 patients. A total of 105 fungal-positive blood culture isolates from 105 patients were identified during the study period with an estimated incidence rate of fungemia of 1.5%. The mean time elapsed from hospital admission to diagnosis was 26.85 days (SD 18.87). The characteristics of the population are shown in Table 1.

The mean age for the entire population was 48 years (SD 19.0), with no significant differences between men and women (50.7 and 42.25 years of age, respectively). Considering the inclusion of patients with different hematologic diagnoses, whose mean age at the onset was different, the behavior of age by diagnosis was explored. 

Regarding the functional status of patients, 78% (82/105) had ECOG 3–4. One third of the fungemias were breakthrough fungemia; 61% of patients had febrile neutropenia, with a mean duration of 21.8 days. The presence of some risk factors was identified in 35% of the patients; most of them had a central venous catheter, with CVC infection in half of them. CVC removal was performed in 40% of patients as an individualized decision, taking into account risk/benefit in the context of patients with neutropenia. The total number of patients who had received HPCT with autologous therapy was 9 (8.6%).

Of the total number of patients, 50 (47.1%) were admitted to the ICU, while the rest were hospitalized on a normal ward. 

Regarding comorbidity, a Charlson index was calculated in all patients with a mean of 3.6, which corresponds to an estimated 10-year survival between 53 and 77%. Overall, 44% (46 patients) had no comorbidity, 56% had some comorbidity, 20% hypertension, 9% heart failure, and 6% diabetes mellitus. Regarding the two groups of interest, in the lymphoma/MM (multiple myeloma) group, more morbidities were identified in relation to the leukemia group, 65% vs. 47%, respectively.

The characteristics by hematological malignancy are shown in Table 2. The HMs were, in order of frequency: acute lymphoid leukemia (ALL) (30%; 32/105), diffuse large B-cell lymphoma (DLBCL) (14%; 15/105), acute myeloid leukemia (AML) (13%), other lymphomas (10%), and multiple myeloma (MM) (8%). Acute leukemias, both lymphoid and myeloid, were the most frequent diagnosis, representing 45% of the observed population, while lymphomas represented 37% of the population. 

### 3.2. Neutrophil Count

The neutrophil count for the entire group of patients had a median of 260 cells/mm^3^, with an IQR of 4540 and an SD of 11.01. 

Two groups of interest were considered given the heterogeneity of the HMs: a group made up of patients with acute and chronic leukemias (ALL, AML, and other leukemias) and another group with lymphoma (diffuse large B-cell lymphoma, follicular, Hodgkin’s, and other lymphomas) and MM.

The median ANC for the leukemia group was 80 cells/mm^3^, while the median for the lymphoma or MM group was 2580 cells/mm^3^. The findings suggest that there is more neutropenia in relation to fungemias in leukemia than in other hematologic neoplasms, such as lymphoma or MM. 

Figure 1 shows the difference in ANC according to the diagnostic group. Some scattered values are observed in both groups.

### 3.3. Microorganisms Identified and Treatment Received

Regarding the identified microorganisms, 98% corresponded to yeasts and 2% corresponded to mycelial fungi. All the mycelial fungi identified corresponded to the genus *Fusarium* spp.; similarly, of the total number of yeasts determined, 82% (86 patients) were *Candida* spp. and 18% were other yeasts, such as *Cryptococcus neoformans* (9%) and *Trichosporum asahii* (3%). Regarding *Candida* spp., 21% corresponded to *Candida albicans*, in contrast with the 61% that were non-albicans *Candida*, with the most frequent species being *Candida tropicalis* (28%), followed by *Candida parapsilosis* (17%), and *Candida krusei* (12%) (Table 3).

Most patients (85%) received effective antifungal treatment, which was administered early (<48 h) in 77% of the patients. The most-used antifungal was an echinocandin (caspofungin) in two thirds of the patients (61%), followed, in order of frequency, by deoxycholate amphotericin B (19%), voriconazole (17%), and fluconazole 16%. At the time of fungemia, 32 patients (30%) had been receiving an antifungal agent for >1 day, and they were classified as breakthrough fungemia. These were secondary to *C. krusei* (10; 33%), *C. parapsilosis* (9; 28%), *C. tropicalis* (9; 28%), *Fusarium* spp. (2; 6%), *Cryptococcus neoformans* (1; 3%), and *C. guilliermondii* (1; 3%). The patients who did not receive antifungal treatment (16 patients), showed a mortality rate of 81%.

Regarding the susceptibility profile of *Candida* spp. fungemia, fluconazole-resistant isolates were found in 9.52% of the patients and resistance to voriconazole was found in 1%. Regarding the *Candida* species, resistance to fluconazole was found in *C. tropicalis* (five patients), *C. krusei* (four patients), and *C. parapsilosis* (one patient).

### 3.4. Survival Analysis

According to the Kaplan–Meier overall survival curve, patients with HM presented a median overall survival of 37 days (Figure 2). The probability of survival at day 30 with a diagnosis of leukemia was 59% (95% CI 46–76), while in the lymphoma/MM group, it was 41% (95% CI 29–58) (*p* = 0.03) (Figure 3). Differences were noted between patients with lymphoma and leukemia. Lymphoma patients were older (56.03 SD 16.5 vs. 40.8 SD 18.2, *p* =< 0.001), more frequently in shock (38.5% vs. 20.8%, *p* = 0.075), and did not have another comorbidity (34.6% vs. 52.8%, *p* = 0.19).

The Cox proportional hazards regression model for overall mortality is described in the univariate analysis (Table 4). The univariate analysis included eight variables of interest for the analysis of global mortality, which led to the following risk factors for the outcomes with statistical significance: age (HR 1.01; *p* = 0.043), diagnosis in the group lymphoma/MM group (HR 1.86; *p* = 0.029), ICU admission (HR 3.14; *p* =< 0.01), and septic shock (HR 1.38; *p* =< 0.01). 

The multivariate analysis showed that diagnosis in the lymphoma/MM group and ICU admission were the risk factor variables for mortality in fungemia with statistical significance (Table 5).

Survival analysis did not differ among different isolated fungi (also considering genera).

## 4. Discussion

Fungal infections represent an emerging problem with a constantly changing epidemiology. Studies on different cohorts of patients with fungemia or candidemia have been published in Latin America and worldwide, observing a change in their epidemiology and etiology, with an increase in non-albicans *Candida* species in HMs, as well as an increase in resistance to antifungals [4,10,11,12,13].

Within the our study population, the majority was under 50 years of age, with a poor performance status (ECOG > 3), however, and without additional comorbidities. This finding is similar to that which has been found in other studies on HMs [3,10,12,14,15,16] in young populations with a poor functional status. 

Regarding the type of HMs, there was a predominance of leukemias with respect to lymphomas, representing 45% of the population, similar to that found in the literature [13,14,16]. Correspondingly, there was a higher frequency of HMs in active treatment (induction, relapse/refractory) [13,14,15].

The incidence of each species of *Candida* spp. varied by geographic area, comorbidity, and demographic characteristics [4,6,17,18,19,20,21]. Although *C. albicans* remains the most common species of fungemia in most studies in North America and northern Europe, there is a growing proportion of non-albicans *Candida* fungal infections with increasing prevalence [3,22,23], with isolates of *C. glabrata* being the most frequent [18,22,24,25]. In contrast, in some published studies from Latin America, non-albicans *Candida* species predominate, and within these, *Candida parapsilosis* and *Candida tropicalis* are the most frequently isolated species [4,6,18]. In our study, most isolates corresponded to yeasts, mainly non-albicans *Candida* (61%), similar to the findings of non-albicans *Candida* predominance previously described in Latin America in the general population, although with a change in its distribution with a higher prevalence of *Candida tropicalis* over *C. parapsilosis* and an increase in the recovery of *C. krusei*. On the other hand, the representation of *C. glabrata* was low (2.8%), in contrast with studies in the Northern Hemisphere and recent studies in Brazil [26,27].

In relation to studies conducted in Colombia, these findings are similar to those described in the cohort study carried out at the INC between 1999 and 2009 [28] with a predominance of candidemia due to *C. tropicalis*, and in contrast with the results of candidemia in patients hospitalized in Bogotá where 26% of patients with HMs were included and whose main microbiological isolation was *C. albicans* with 66.4% [29]. This could suggest a relationship between patients with candidemia due to *C. tropicalis* in the context of HMs, in accordance with the risk factors described for candidemia due to *C. tropicalis* (such as ALL and neutropenia), and prolonged stays in the ICU [14,30,31,32,33,34], characteristics found in the population of our study.

There is less information regarding mycelial fungi, although an increase in their prevalence in cancer centers has been described [35,36]. The main culprit in the context of fungemia is *Fusarium* spp., consistent with this cohort, which is associated with a worse prognosis [13,37,38].

In different publications on cancer patients who develop fungemia, neutropenia has been widely recognized as a risk factor, with ANC and its duration representing a higher risk [16,23,33]. In our cohort, neutropenia was present in 61% of patients, in a prolonged form (mean: 22 days) and with an ANC < 500.

CVC removal in patients with fungemia has been widely discussed in the literature [2,17,39,40,41,42]. In this cohort, 40% of the fungemias were related to this device, which was removed in half of the cases. CVC removal is seen as an individualized decision that considers the probability of it being a source of persistent infection, in contrast with other specific procedure-related risks in this group of patients. There are no clinical trials that have evaluated CVC removal [43], and retrospective studies have had divergent results [41]. However, some studies suggest a decrease in mortality and greater clinical success associated with its withdrawal [2,15,40,42,43,44].

As mortality higher than 60% has been reported in patients without antifungal treatment [45], recent studies suggest that the early initiation of antifungal treatment has been related to decreased mortality, both early and late [2,44,46,47], as well as the use of echinocandins as a first-line treatment having an impact on mortality [2]. In our study, 85% of the patients received antifungal treatment, mostly initiated early and with the use of echinocandin being more frequent. 

Consistent with reported cohorts [3,48], breakthrough fungemia was found in one-third of patients, the majority being due to non-albicans *Candida* infections, in accordance with that which has been described in other studies [3,16,49].

The decreased susceptibility of non-albicans species to azoles and echinocandins has been reported [50,51,52]. The proportion of isolates with resistance to fluconazole was 9%, similar to the findings of previous studies that reported a low proportion of resistance in isolates from *C. albicans, C. parapsilosis*, and *C. tropicalis*, unlike global resistance in *C. krusei* and increasing resistance in *C. glabrata* [4,7,53,54,55,56,57].

In different published studies, global mortality due to fungemia in HMs at 30 days has been described as being between 30% and 60% [3,10,12,13,14,15,16,42]. Our study had similar results, with an overall mortality rate of 50%, in accordance with that which has been described in the INC cohort, whose reported mortality was 52.4% [28]. The survival probability at day 30 for patients diagnosed with leukemia was 59% (95% CI 46–76), and in the lymphoma/MM group it was 41% (95% CI 29–58) (*p* = 0.03).

In the univariate analysis, age (HR 1.01; *p* = 0.043), diagnosis in the lymphoma/MM group (HR 1.86; *p* = 0.029), ICU admission (HR 3.14; *p* =< 0.01), and septic shock (HR 1.38; *p* =< 0.01) were statistically associated with increased mortality. However, in the multivariate analysis, the diagnostic group (HR 1.72; *p* = 0.04) and ICU admission (HR 3.08; *p* =< 0.01) were the variables that maintained their association with statistical significance as risk factors for mortality in patients with fungemia. Septic shock and ICU admission have been widely described as a risk factors associated with mortality [10,14,15,33,50,58]; however, there is little information in the literature regarding the study of mortality according to the type of hematologic neoplasia [34].

Interestingly, advanced age, the presence of organ dysfunction, kidney failure, and neutropenia were not related to mortality, as previously described in different studies [2,10,14,58,59]. Similarly, protective factors for mortality, such as antifungal prophylaxis and remission of oncological pathology as described in other studies, were not found [3]. Finally, no association was found between mortality and gender or species, as described in the literature [11,24,34].

Among the strengths of the study, it can be highlighted that it is the first multicenter study of fungemia in hematologic neoplasms carried out in Colombia. Despite being a retrospective study, the proposed variables were found in most patients, and breakthrough fungemia were present, which is related to the immunosuppression degree due to the underlying disease and the treatments administered.

This study has several limitations: firstly, it has a retrospective nature due to information bias, given that the data were collected from clinical histories; secondly, there is an absence of some data, mainly in the ECOG and the susceptibility test variables; thirdly, there are uncontrollable effects, such as comorbidities and the hematologic neoplasia status, which can affect the mortality outcome; and fourthly, the incidence of fungemias and specific species identified might have been impacted by local fungal epidemiology—therefore, their findings may not be generalizable to other institutions.

In conclusion, fungemia is a relevant problem in patients with HMs. Due to their high morbimortality, it is important to know the epidemiological profile of these patients and the factors related to mortality to implement prevention diagnosis and timely treatment strategies, which are key to reducing the burden of fungal diseases. 

## Figures and Tables

**Figure 1 jof-09-00400-f001:**
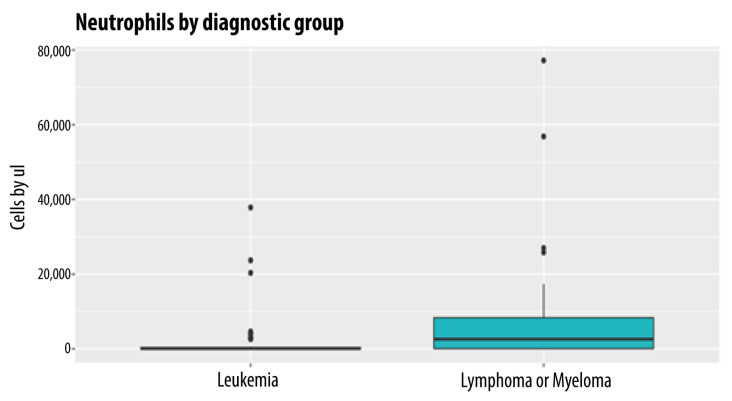
Neutrophil count by diagnostic group.

**Figure 2 jof-09-00400-f002:**
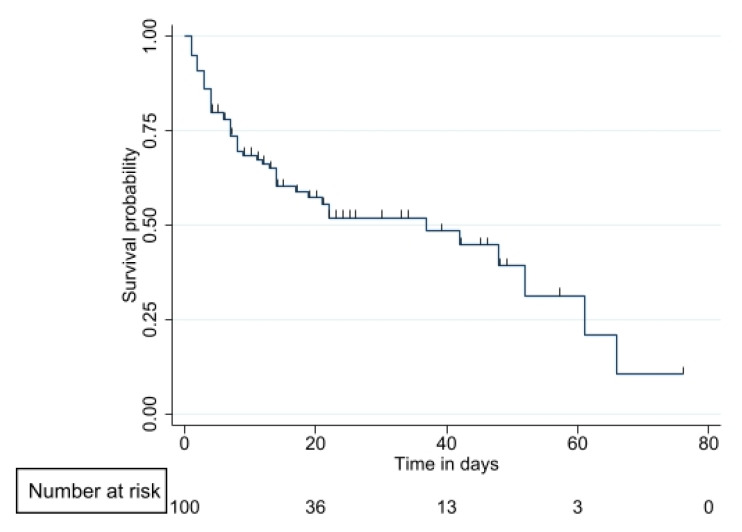
Kaplan–Meier overall survival curve.

**Figure 3 jof-09-00400-f003:**
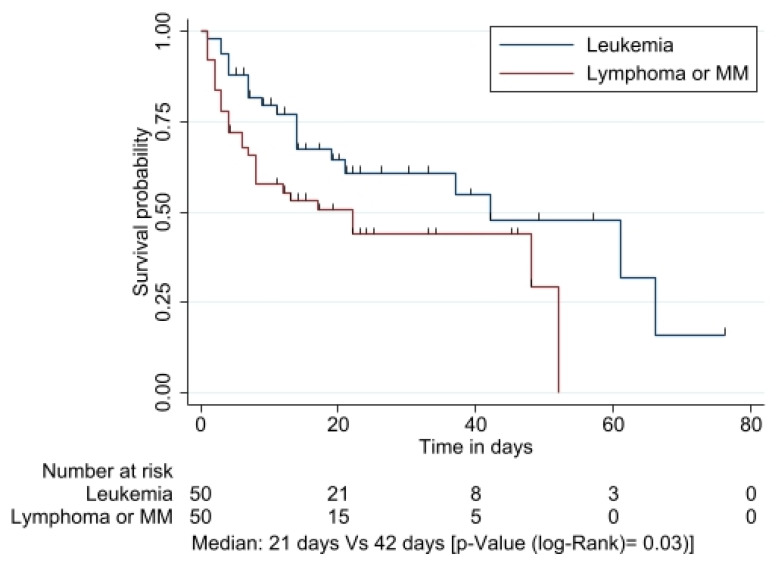
Kaplan–Meier overall survival curve by HM group.

**Table 1 jof-09-00400-t001:** Clinical characteristics and risk factors.

Variable	Total n (%)
Patients included	105
Gender = female (%)	55 (52.4)
Age = mean (SD)	48.38 (19.03)
Diagnostic stage of hematologic malignancy (%)	
Active (new diagnosis/induction)	54 (51.5)
Consolidation	3 (2.9)
Relapse/refractory	42 (40.0)
Remission	6 (5.7)
Radiation therapy = yes (%)	3 (2.9)
ECOG (%)	
0–2	10 (9.5)
3	43 (41.0)
4	39 (37.1)
No data	13 (12.4)
Antifungal prophylaxis = yes (%)	32 (33.3)
ANC = mean (SD)	260 (11.01)
FN = yes (%)	64 (61.0)
Duration of neutropenia days = mean (SD)	21.78 (16.13)
Previous episodes of FN = yes (%)	29 (27.6)
Risk factor = yes (%)	37 (35.2)
ANC < 500 = yes (%)	57 (54.3)
HPCT = yes (%)	9 (8.6)
Corticosteroids = yes (%)	13 (12.4)
Cyclosporine = yes (%)	10 (9.5)
Severe congenital immunodeficiency = yes (%)	1 (1.0)
CVC = yes (%)	70 (66.7)
Parenteral nutrition = yes (%)	5 (4.8)
Mucositis = yes (%)	17 916.2)

ECOG: Eastern Cooperative Oncology Group; ANC: absolute neutrophile count; FN: febrile neutropenia; HPCT: hematopoietic progenitor cell transplant; CVC: central venous catheter.

**Table 2 jof-09-00400-t002:** Characteristics by hematological malignancy.

Disease	n (%)	Age Mean (SD)	Patients with Neutropenia < 500 n(%)	Neutrophil Count Median (IQR)	Days of Neutropenia Median (IQR)	Hospital Stay (Days) Median (IQR)	ICU Stay (Days) Median (IQR)	Non-Survivors n (%)	ECOG Median
ALL	32 (30.4)	34.34 (14.64)	26 (81.2)	95 (347.5)	21.5 (19.5)	44 (29.5)	8 (16.75)	14 (43.7)	3
DLBCL	15 (14.2)	57.46 (14.91)	2 (20)	6130 (10,735)	9 (8)	39 (25)	10.5 (9.25)	9 (60)	4
AML	14 (13.3)	48.64 (20.26)	12 (100)	25 (219.75)	34 (19.25)	43.5 (21.5)	3.5 (3.75)	6 (42.8)	3
Other lymphoma	11 (10.5)	61.27 (15.60)	4 (36.3)	2950 (6474)	10.5 (7.5)	41 (38.5)	7.5 (8)	3 (27.2)	3
Multiple myeloma	8 (7.6)	64.12 (9.38)	0	4845 (4664)	–	27.5 (41.5)	18 (13)	4 (50)	4
Hodgkin’s lymphoma	6 (5.7)	53 (19.35)	2 (33.3)	785 (5605)	14 (6)	56 (33.5)	16 (29.5)	4 (66.6)	4
PTCL	5 (4.7)	53.2 (19.57)	2 (40)	10,490 (8130)	17 (0)	19 (7)	3 (3)	5 (100)	4
Follicular lymphoma	3 (2.8)	49.11 (18.87)	1 (33.3)	740 (542)	7.5 (7.5)	28 (15.5)	19.5 (2.5)	2 (66.6)	3
CLL	2 (1.9)	75.5 (4.94)	2 (100)	15 (5)	20.5 (1.5)	45 (23)	3 (0)	2 (100)	4
CML	2 (1.9)	55 (8.48)	1 (50)	18,965 (18,885)	10 (10)	40 (2)	16 (0)	1 (50)	2
APL	2 (1.9)	49.2 (12.02)	0	12,230 (8120)	–	10.2 (44)	33.5 (8.5)	1 (50)	3
Burkitt’s lymphoma	1 (0.9)	32	0	10	–	39	5	0	3
Malignant hystyocytosis	1(0.9)	27	1(100)	10	31	93	0	1 (100)	3
Lymphoblastic lymphoma	1(0.9)	21	1 (100)	2	13	43	–	0	3
Mycosis fungoides	1(0.9)	53	0	8290	–	84	–	0	4
Other leukemia	1(0.9)	26	0	200	–	34	11	1 (100)	4

ECOG: Eastern Cooperative Oncology Group; ANC: absolute neutrophile count; ICU: intensive care unit; ALL: acute lymphoid leukemia; DLBCL: diffuse large B-cell lymphoma; AML: acute myeloid leukemia; PTCL: peripheral T-cell lymphoma; CLL: chronic lymphocytic leukemia; CML: chronic myeloid leukemia; APL: acute promyelocytic leukemia.

**Table 3 jof-09-00400-t003:** Identified microorganisms.

	Microorganism	n (%)
1	*Candida tropicalis*	29 (28)
2	*Candida albicans*	22 (21)
3	*Candida parapsilosis*	18 (17)
4	*Candida krusei*	13 (12)
5	*Cryptococcus neoformans*	9 (8.5)
6	*Candida glabrata*	3 (2.8)
7	*Trichosporum asahii*	3 (2.8)
8	*Fusarium* spp.	2 (1.9)
9	*Candida guillermondii*, *Cryptococcus laurentii*, *Geotrichum* spp., *Malassezia pachydermatis*, *Rhodotorula mucilaginosa*, *Trichosporum beigelii*	1 (1)

**Table 4 jof-09-00400-t004:** Univariate analysis for overall mortality.

Variable	HR	IC 95%	*p*-Value
Age	1.01	1–1.03	0.04 *
Gender	0.67	0.39–1.17	0.16
Diagnosis by groups: lymphoma/MM	1.91	1.08–3.36	0.02 *
ECOG	1	0.99–1	0.83
ICU admission	3.14	1.74–5.67	<0.001 **
ANC < 500	0.55	0.31–0.95	0.03
Septic shock	2.6	1.5–4.52	<0.001 **
CVC	1.38	0.76–2.51	0.27
Relapsed/refractory clinical status	0.71	0.4–2.27	0.25

MM: multiple myeloma; ECOG: Eastern Cooperative Oncology Group; ICU: intensive care unit; ANC: absolute neutrophil count; CVC: central venous catheter. ** major statistical significance; * minor statistical significance.

**Table 5 jof-09-00400-t005:** Multivariate analysis for overall mortality.

Variable	Coefficient	HR	z	*p*-Value
Diagnosis by groups: lymphoma/MM	0.583	1.792	2.039	0.04 *
ICU admission	1.128	3.089	3.749	<0.001 **

MM: multiple myeloma; ICU: Intensive Care Unit. ** major statistical significance; * minor statistical significance.

## Data Availability

The data presented in this study are available on request from the corresponding author. The data are not publicly available due to IRB (Institutional Review Board).

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
