# Peer review of "Fungemia in Hospitalized Adult Patients with Hematological Malignancies: Epidemiology and Risk Factors"

_jof, 2023, doi:10.3390/jof9040400_

Round 1

Reviewer 1 Report

This is a retrospective study carried out during 7 years (2012-2019) in 5 hospitals in Bogota (Colombia) about fungemia in adults with hematological malignancies. The work describes the epidemiology of the cohort and analyzes aspects related to mortality. The manuscript is in line with other similar works, it is well prepared and presented but does not offer very significant and novel aspects.

We recommend reviewing the following aspects:

- The authors define severe neutropenia at neutrophil counts < 500 cells/ul; however, in the literature the term severe neutropenia is reserved for neutrophil counts <100 cell/ul, review.

- The authors mention that early antifungal therapy is initiated <48h after microbiological isolation. This makes it necessary to explain what is considered microbiological isolation: identification of yeast by gram-stain in blood?, identification of fungal species?, sensitivity report?. Early microbiological detection techniques were applied?, Malditof...

- The authors mention sensitivity studies, even in the discussion it is mentioned that the frequency of isolates resistant to fluconazole was 9%. However, there is no mention of the methodology used in sensitivity, micro dilution?, Sensititre?, CLSI?, EUCAST?, nor the cut-off points used.

- 33% of antifungal prophylaxis was mentioned in the series. Information should be provided on this aspect: criteria for antifungal prophylaxis, type of antifungal and type of patients in which it was used.

- More information is needed on the types of malignancies, including another table; the data in the text do not add up.

- 30% of breakthrough fungemia is reported. This aspect is very significant.  In a work focused on candidemia, more precise information should be given on the management and removal of catheters. In the discussion (not in the results) it is mentioned that there were 40% of bloodstream catheter-related candidemia; however, catheter removal only occurred in 50%. This deserves a more detailed explanation on the use of diagnostic techniques, differential blood cultures or justification for catheter withdrawing or not.

- The study does not provide extensive information on patient comorbidity, including catheter management. Some factors could form part of the risk factors. The high mortality rate (50%) and the fact that the ECOG was >3 in 83% of the patients is striking. It speaks of a very fragile population, with other clinical aspects more complex than those associated with simple bloodstream catheter-related candidemia.

- 19% received amphotericin B. Were lipid amphotericins used? or was it deoxycholate amphotericin B?

- Finally, in the mortality analysis, the 2 independent associated factors were admission to the ICU and having lymphoma/myeloma versus acute leukemia. This finding is contrary to other series where mortality is higher in patients with acute leukemia. The authors should give more explanations about this remarkable fact or provide more information about the oncological treatment applied to the patients.

- Table 3 presents overlapping data and does not read well. From an statistic point of view, we believe it would be advisable to present the data with HR, CI and p. This is more usual than the coefficient or Z value.

Author Response

Dear reviewer 1,

We appreciate the effort to review the article and thank you for all the suggestions. Please see the attachment, we respond to each of the comments.

Kind regards
Dr. Vargas Espindola

Reviewer 2 Report

In this retrospective cohort study in 5 hospitals in Colombia from 2012 to 2019, the study authors report on the epidemiological and clinical characteristics and antifungal susceptibility profiles of hematologic malignancy patients with fungemias. They reported 105 patients with predominantly non-Candida albicans fungemias. ICU admission and a diagnosis of lymphoma/multiple myeloma were associated with mortality on multi-variate analysis.

Overall, the study is well designed and provides important information on the emergence of non-Candida albicans fungemias in these immunocompromised patient populations. I have some comments to add that would hopefully increase the value provided by the study data.

Major Comments

1. The study authors report 105 fungal positive blood cultures being identified in 105 patients during their study period. It would be educative if they provided the total number of charts reviewed, so that one may be able to infer an incidence rate of invasive fungal infections (“xx/xx patients, xx%”).

2. The study authors report that they had 47 patients (44.8%) of patients in active stage. Does that mean new diagnosis, but not treated? It would be helpful to provide the exact information. It would also be helpful if they included the definitions of the diagnostic stages in the Materials and Methods section. In case certain national/international staging guidelines were followed, that should be included as well.

3. The study authors noted that 32 patients (33.3%) of their cohort were on antifungal prophylaxis. There are some unanswered questions in this regard – what was the basis of antifungal prophylaxis (institution protocol, physician preference?), which antifungal was used, what was the median duration of antifungal usage? This becomes particularly important with the increasingly reported incidence of breakthrough invasive fungal infections.

4. The study authors briefly mention the susceptibility profile of Candida spp. in their patients with fungemias (in page 6 of 14), with fluconazole-resistant isolates seen in 9.52% and voriconazole resistance in 1%. It would be helpful if they mentioned whether this was related to a particular Candida species (such as Candida krusei and Candida glabrata that are known to have emerging antifungal resistance). If needed, they should consider including a Supplementary table in this regard.

5. In Table 3, the study authors documented certain variables that they included in the univariate analysis for overall mortality. Considering that a large proportion of their population (42 patients, 40%) were in relapsed/refractory clinical status, did they also consider that as a variable to consider for mortality analysis? This would be an important clinical point, as it would raise awareness amongst clinicians to consider that as a risk factor for fungemias while taking care of these critically ill patients.

6. In their study limitations, the study authors should also consider adding that their incidence of fungemias and specific species identified might have been impacted by local fungal epidemiology – their findings may thus not be generalized to other institutions. This is (understandably) a limitation that is seen with a lot of studies related to invasive fungal infections, but it would be important to be highlighted.

Minor Comments

1. Page 1 of 14, line 45-46 – please provide a reference for the statement “Yeasts are the leading cause of fungemia, with Candida spp. as the most common agent with a reported incidence of 0.23 – 1.5% in hospitalized patients”

2. Page 2 of 14, line 68 – the study authors should consider rephrasing “presented fungemia” as “presented with fungemias”

3. Page 2 of 14, line 94 – the study authors should consider rephrasing “mold non Aspergillus” with “mold (non Aspergillus) species”

4. Page 2 of 14, line 99 – please clarify whether the statement “or on catheter-tip culture” would indicate growth of the fungus on catheter tip culture, or if it is related to the prior statement about growth in blood culture from line occurring faster than in peripheral blood.

5. Page 3 of 14, lines 132-134 – it is unclear whether the following lines need to be included in the Ethical aspects section “The Materials and Methods should be described with sufficient details to allow others to replicate and build on the published results”. Please check the journal requirements. Please mention of this needs to be included as part of their institutional/national protocols.

6. Page 4 of 14, Table 1 – what does “Si” refer to, when associated with ANC<500 and HPCT? Please clarify that.

7. Page 4 of 14, Table 1 – with patients who had received HPCT, what type of therapy did they receive (autologous/allogeneic/CAR-T)? It would be very educative to include that information in the Results section.

8. Page 5 of 14, line 194 – the study authors report in line 194 that the incidence of Candida tropicalis was 28%. This is also mentioned in the Abstract. However, in the accompanying Table 2 on the same page, it is mentioned as 27%. Please clarify this.

Author Response

Dear reviewer 2,

We appreciate the effort to review the article and thank you for all the suggestions. Please see the attached file, we respond to each of the comments.

Kind regards,
Dr. Vargas Espindola

Reviewer 3 Report

To the authors, 

Espíndola et al. conducted an interesting study titled “Fungemia in malignancies haematologic in a middle income country”. The retrospectively described patients with hemstologic maligniancies who got diagnosed with fungemia. They described the baseline characteristics as well as the outcomes in this cohort.

Comments:

1.     The title should be rephrased.

2.     Extensive language editing is required.

3.     Why did exclude patients with HIV? Please explain. Moreover, how many patients with HIV were excluded? A flow diagram could help. 

4.     Line 94: What were these “symptoms of systemic fungal infection?”

5.     Why did you exclude aspergillus?

6.     How many patients were at a normal ward or an ICU?

7.     How high was the mortality in the 15% who did not receive antifungal treatment?

8.     Can you provide the time from hospital admission to diagnosis?

9.     The biggest problem is the lack of a control group which makes it difficult to infer the relevance and impact of the fungemia in this population.

Author Response

Dear reviewer 3,

We appreciate the effort to review the article and thank you for all the suggestions. Please see the attached file, we respond to each of the comments.

Kind regards
Dr. Vargas Espindola

Round 2

Reviewer 1 Report

The work has been clearly improved and has satisfactorily incorporated the recommendations that were made. The methodology has been rewritten with greater precision and the results and tables are more accurate.

Reviewer 3 Report

Thank you. No further comments from my side.